# The effect of diaphragmatic breathing and diaphragmatic mobilization on physical performance, fear of falling, and quality of life in community-dwelling older adults: A randomized controlled trial

Soyba Nazir[1], Witaya Mathiyakom[2], Muhammad Awais Tassawar[1], Anong Tantisuwat[1]*

1 Faculty of Allied Health Sciences, Chulalongkorn University, Pathumwan, Bangkok, Thailand,
2 Department of Physical Therapy, California State University, Northridge, California, United States of America

* anong.ta@chula.ac.th

## Abstract

### Background

Falls are a significant health concern among older adults, leading to reduced mobility, fear of falling, and poor quality of life. Diaphragmatic breathing (DB) and diaphragmatic mobilization (DM) may serve as therapeutic interventions to address these issues.

### Objective

This study evaluated the effects of DB and combined DB and DM (DB+DM) on physical performance, fear of falling, and quality of life in community-dwelling older adults.

### Methods

Fifty-four older adults (65–75 years) were randomized into group 1, DB; group 2, DB+DM; and group 3, control groups (18 per group). Interventions were delivered twice weekly for 8 weeks. Outcomes, including balance, gait performance, lower extremity strength, fear of falling (FoF), fatigue, and quality of life (QoL), were measured at baseline, post-treatment (8th week), and follow-up (10th week).

### Results

Significant group-by-time interaction effects (p < 0.05) were observed in all outcome measures except a few domains of Short-form 36 (SF-36). At post-treatment, the mini-BEST scores and the time to complete 5-time sit-to-stand test (5xSTS) of the DB and DB+DM groups did not significantly increase compared to baseline. The

**Data availability statement:** All relevant data are within the manuscript and its Supporting Information files.

**Funding:** The author(s) received no specific funding for this work.

**Competing interests:** NO authors have competing interests.

timed up-and-go (TUG) scores ($\eta^2 = 0.279$) and gait velocity (GV) ($\eta^2 = 0.619$) of both interventions significantly improved ($p < 0.05$) from baseline to post-treatment and follow-up. Scores for active balance confidence (ABC) ($\eta^2 = 0.706$), fatigue severity scale (FSS) ($\eta^2 = 0.584$), and specific domains of SF-36, physical function ($\eta^2 = 0.211$), pain ($\eta^2 = 0.173$), and general health ($\eta^2 = 0.168$) showed significant improvements in both intervention groups compared to the control group.

## Conclusion

DB and DB + DM interventions significantly improved gait performance, FoF, fatigue, and QoL in community-dwelling older adults. However, their therapeutic effects on balance were limited and need further investigation.

## Introduction

Falls are a significant health concern for older adults, often resulting in serious injuries such as hip fractures and head trauma, which can lead to impaired mobility, early admission into long-term care facilities, and even death [1]. At least one in three older adults over 65 experiences a fall annually [2–4]. Falls increase the likelihood of subsequent falls and significantly diminish the quality of life (QoL) [5,6]. Furthermore, the negative consequences associated with falls encompass both physical and psychological issues, including fear of falling (FoF), which, in turn, elevates the risk of injury and lower health-related QoL [7]. Therefore, a comprehensive fall prevention and rehabilitation program must address physiological factors such as balance and postural control and the associated biopsychosocial factors for reducing fall risk and improving QoL.

Balance and postural control are essential for maintaining stability and preventing falls, yet both decline progressively with age [8,9] Spinal alignment, particularly the lumbopelvic segment, is crucial for maintaining proper spinal function as it stabilizes the upper body over the lower extremities and supports an upright posture [10]. However, age-related changes, such as intervertebral disc degeneration, reduced muscle strength, and connective tissue fragility, can impair spinal alignment, leading to disruption in posture and balance [11–13]. These age-related impairments in spinal alignment and postural control heighten fall risk and emphasize the importance of exploring interventions that target balance and fall prevention in older adults.

The diaphragm, commonly recognized for its role in respiration, also plays an integral part in postural stability [14,15]. It works with abdominal muscles to create a hydraulic effect, increasing intra-abdominal pressure [16,17], which stabilizes the lumbar spine and enhances balance [9,15]. Reduced diaphragm movement during breathing compromises proprioceptive abilities, leading to diminished sensory input necessary for postural control and balance [18]. This highlights the potential for therapeutic interventions, such as diaphragmatic breathing (DB), to improve both respiratory and postural functions, thereby reducing fall risk and enhancing QoL.

Previous studies suggest that DB provides multiple benefits, such as improved respiratory muscle strength, motor function, diaphragm proprioception, and lumbar stability, collectively enhancing postural stability and balance [19,20]. DB exercises have been shown to improve spinal mobility in young adults [21], postural stability in adults with lumbar instability [20], and balance in athletes with chronic low back pain [22]. Eight weeks of DB exercises have been associated with reduced balance errors and improved static and dynamic balance in the young population [20]. However, its application in older adults, particularly in the context of fall prevention, remains underexplored. Findings from younger adults cannot be assumed to generalize to older populations due to age-related physiological and functional differences. This gap in research presents an opportunity to investigate whether DB could similarly benefit older adults by improving their balance and reducing fall-related risks.

Diaphragmatic mobilization (DM) is a manual therapy technique [23] that complements DB by enhancing thoracic and spinal mobility and respiratory function [24], essential for postural control [25]. DM techniques have been applied to manage musculoskeletal conditions such as reduced lumbar and chest wall flexibility [26], shoulder pain, and rotator cuff injuries [27], demonstrating improvements in spinal and shoulder mobility and QoL in young adults [28]. Furthermore, DM techniques affect respiratory muscle strength in obese individuals [29], and enhanced respiratory muscle strength leads to improved static balance and prevention of falls in older adults [30]. Despite these promising findings, the combined effects of DB and DM on balance and other physical and psychological outcomes, such as FoF, have yet to be investigated, highlighting the need to address both physiological and psychological factors in fall prevention strategies.

FoF is one of the psychological factors that significantly impacts the physical and psychological well-being of older adults [31]. FoF, defined as a low perceived self-efficacy in preventing falls during daily activities [32], is both a cause and a consequence of falls [33]. Among community-dwelling older adults, the prevalence of FoF was 50.1% and 71.4% for non-fallers and fallers, respectively [34]. FoF is associated with depression [35], an altered gait [36], limitations in activities of daily living [35,37]. increased fall risk [37], and declined QoL [7]. FoF has a strong connection to physical and psychological health outcomes, emphasizing the importance of interventions that address FoF alongside balance and other functional impairments.

This study aimed to investigate the DB and combined DB and DM (DB + DM) on physical and psychosocial outcomes in older adults. We hypothesized that this non-invasive approach would significantly enhance balance, gait velocity, lower extremity strength, and quality of life while reducing the fear of falling and fatigue. The findings from this study may help identify the extent to which DB and DB + DM exercises improve functional outcomes relevant to fall risk in older adults.

## Methodology

### Study design

This randomized controlled trial investigated the effects of DB and combined DB + DM on physical performance, FoF, and QoL in community-dwelling older adults. The study employed a single-blind design, in which the outcome assessor was blind to group allocation. Participants were recruited from senior living communities (Maa Basaira, Bait-ul-Zaif, Afiat, Pakistan) and a physiotherapy clinic (Mobility Care, Pakistan). The study protocol was approved by the Ethics Committee of Chulalongkorn University (COA No. 086/67), and all participants provided written informed consent prior to participating in the study. The trial was registered with the Thai Clinical Trial Registry (TCTR20240520003). Recruitment of participants took place from May 2024 to June 2024, and data collection, including intervention and follow-up assessments, was completed by December 2024.

### Participants

Community-dwelling older adults aged 65–75 years were eligible if they had a mini balance evaluation system test (mini-BEST) score <16, lower extremity muscle strength ≥ grade 3 (manual muscle testing), could walk short distances

independently without assistive devices, and had normal or corrected vision. Exclusion criteria included pre-existing diaphragmatic breathing patterns, participation in balance training or aerobic activities (e.g., yoga), neurological or musculoskeletal conditions affecting gait and balance, use of medications interfering with postural balance, pelvic muscle dysfunction (≥3 items on the Cozean pelvic health screening questionnaire) [38], positive diastasis recti (>2 mm inter-recti distance), and BMI > 30 kg/m$^2$.

## Sample size and randomization

The sample size was estimated using G*Power (version 3.1.9.4) for a repeated measures ANOVA (within-between interaction), with a standardized effect size of 0.3 (Cohen's *f*), a significance level of 0.05, a power of 0.8, and an attrition rate of 30% observed in the pilot study, resulting in an overall sample size of 39 participants. The recruitment target was increased to ensure robustness and account for potential unforeseen challenges. After screening 60 individuals, 54 participants meeting the inclusion criteria were enrolled. These participants were randomly assigned based on a computer-generated sequence into three groups (18 per group): DB, DB + DM, and Control as shown in Fig 1. The allocation of the subject group was concealed using opaque envelopes by an independent researcher.

## Treatment protocols

**DB Group:** In this group, participants performed DB for 20 minutes per session [20], in three 6-minute sets [39] with a 1-minute rest between sets, twice a week for 8 weeks. Participants were instructed to place one hand on the chest and

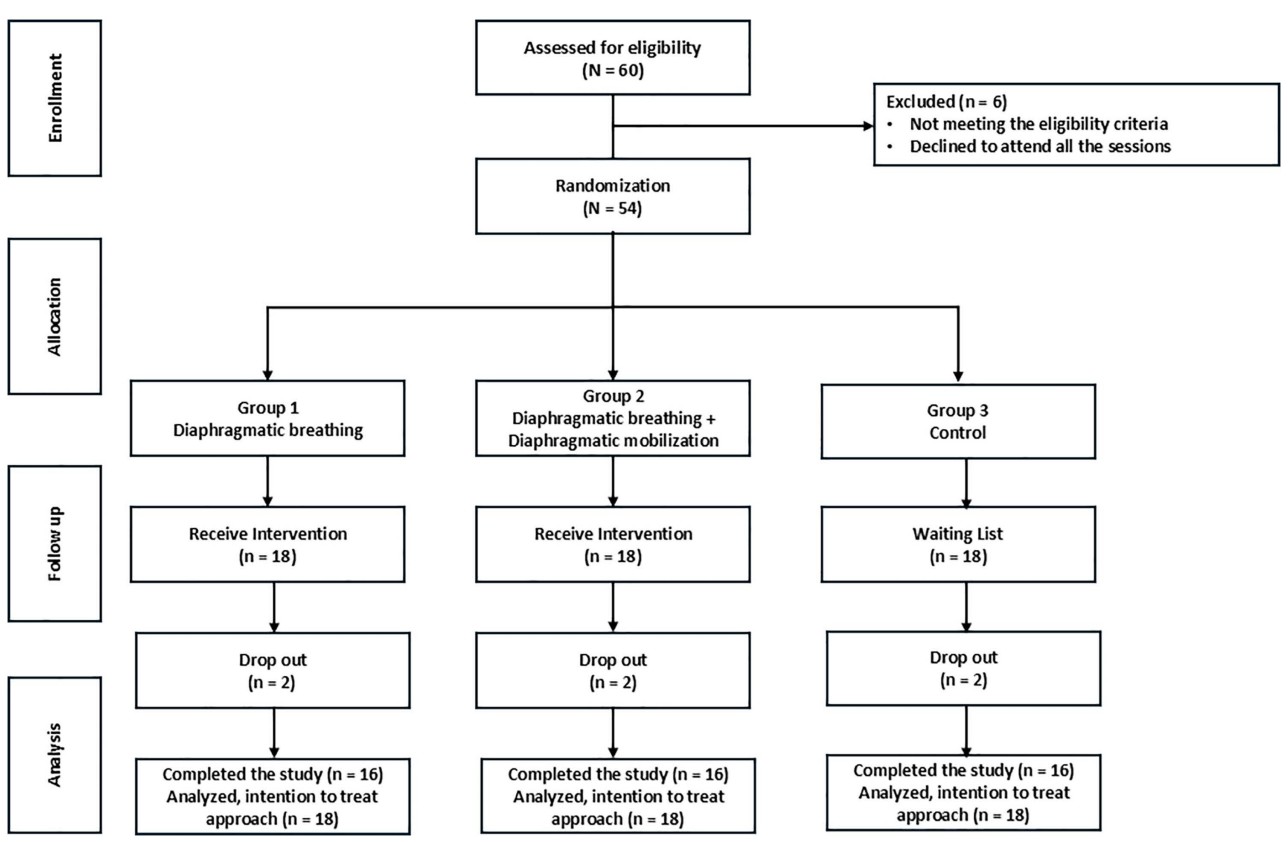

**Fig 1. A Consort diagram.**

the other on the abdomen, ensuring minimal chest movement. They were asked to breathe slowly and deeply through the nose for approximately 4 seconds and exhale through the mouth for 6 seconds [28]. The focus was on expanding the abdomen while keeping the chest as still as possible. **DB+DM Group:** Participants received a combined intervention of DB and DM for 20 minutes per session, twice a week for 8 weeks. Each session included three 2-minute sets of DB followed by three 3-minute sets of DM, with 1-minute rest between sets. During DM, participants lay in a supine position while the physiotherapist applied mobilization by positioning their thumbs on the xiphoid process, grasping the costae, and moving their fingers in a figure-8 pattern to enhance relaxation and diaphragmatic movement [28]. The breathing pattern was synchronized with the mobilization, 4-sec inhalation, and 6-sec exhalation. [28], and a DB progression protocol was followed for both treatment groups. The time sequence for both intervention groups is shown on Fig 2. **Control Group:** Participants were on a waiting list and received no intervention during the study period. However, they underwent the same assessment schedule as the intervention groups. To minimize potential attention bias, control group participants received the same amount of contact during assessment visits and were provided with general health and safety information that was unrelated to the intervention content.

**Progression protocol for DB.** Progressive overload was implemented to promote neural and muscular adaptations for increased training capacity. The DB progression sequence was as follows:

- **Week 1–2:** Supine and Crocodile breathing respectively.
- **Week 3–4:** Supine and Crocodile breathing with TheraBand, respectively.
- **Week 5–6:** Seated and 90/90/90 breathing, respectively.
- **Week 7–8:** Seated and 90/90/90 breathing with TheraBand, respectively.

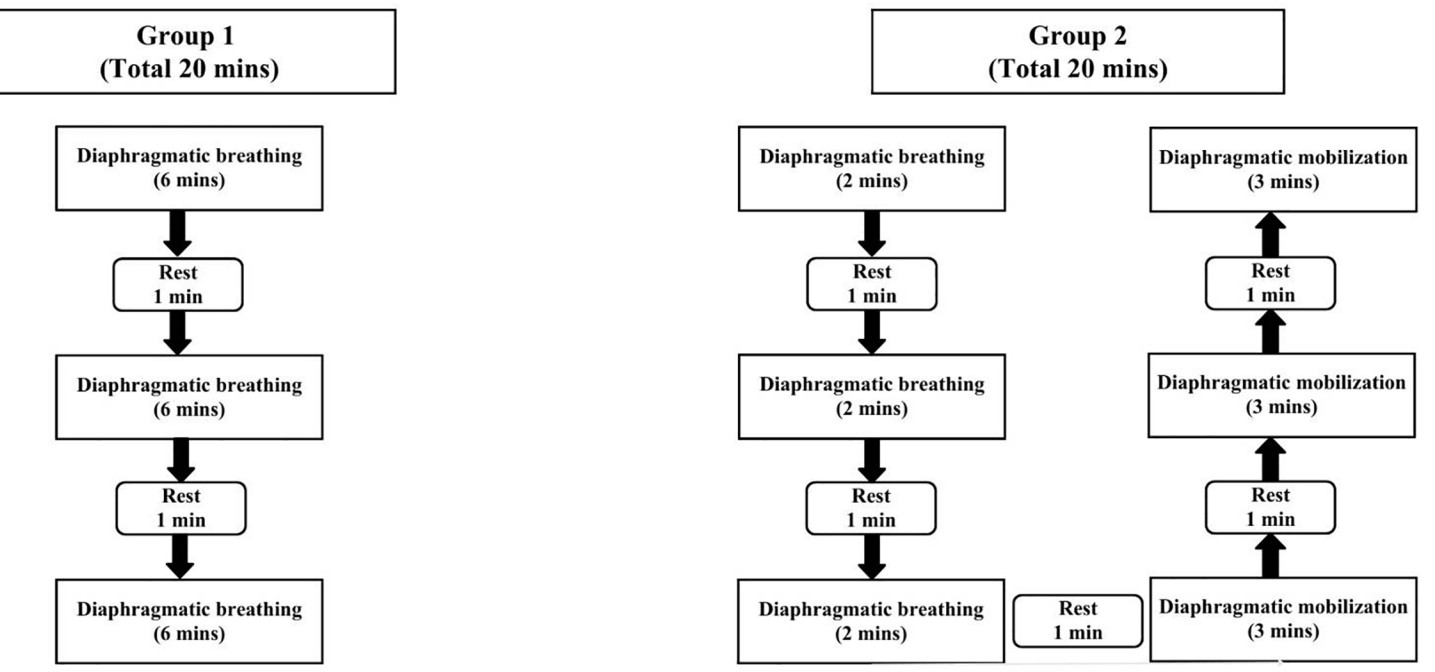

**Fig 2. The time sequence for both intervention groups.**

For resistance, a standard yellow TheraBand® (1.3 kg resistance) was secured around the lower ribcage and dia-phragm [20,22,40], providing consistent resistance across all positions to ensure intervention uniformity as shown in Fig 3.

### Monitoring and safety

Blood pressure (BP) was monitored before and after each session in both intervention groups to ensure cardiovascular safety. After each set, the rate of perceived exertion (RPE) was assessed using the Borg scale. A one-minute rest between sets was implemented to maintain moderate exercise intensity (11–13). These measures ensured participant safety and adherence to the prescribed intensity throughout the intervention.

### Outcome measures

The primary outcome of this study was postural stability or balance, assessed using the mini-BEST test, which had excellent inter-rater reliability ICC = 0.992 [41]. Secondary outcomes included gait performance, measured by the Timed Up and Go (TUG) test and gait velocity (GV); lower extremity strength (LES), measured by the Five-Times Sit to Stand Test (5xSTS); FoF, measured by the Activities-specific Balance Confidence (ABC) Scale; overall fatigue, assessed by the Fatigue Severity Scale (FSS); QoL, evaluated using the Short Form-36 (SF-36). The SF-36 comprises eight domains, including physical function (PF), role limitations due to physical health (RLP), role limitations due to emotional health (RLE), energy, mental health (MH), social function (SF), pain, and general health (GH), all of which were computed accordingly [42]. Measurements were taken at three time points: baseline (pre-intervention), post-intervention (8th week), and follow-up (10th week).

### Statistical analysis

Data were analyzed using IBM SPSS Statistics version 29.0.1.0 (IBM Corp., Armonk, NY, USA). Descriptive statistics (mean, SD) were calculated for all outcome measures. Mixed ANOVA evaluated intervention effects on all outcome measures, with time (baseline, post-treatment, follow-up) as a within-subject factor and group (DB, DB + DM, control) as a between-subject factor. Adherence was monitored throughout the 8-week intervention. Participants who missed more than two sessions were required to complete makeup sessions. All participants who remained in the study met the adher-ence requirement, resulting in a 100% adherence rate among completers. An intention-to-treat (ITT) approach was used, including the data of all 54 randomized participants (n = 18 per group) in the analysis. Although 6 participants (2 per group) did not complete the study, their missing data were managed using the last observation carried forward (LOCF) method. This method was applied only for the six missing follow-up values to preserve the intention-to-treat sample. Post hoc pair-wise comparisons with Bonferroni corrections identified pairwise differences. Statistical significance was set at 0.05.

## Results

S1 Table summarizes the demographic characteristics of all subjects. There was no statistically significant difference in all demographic characteristics among the three groups (p > 0.05), indicating similar prognostic factors at the beginning of the study. S2 Table provides the overall statistical analysis results of all outcome measures. Overall, significant group-by-time interaction was observed for all outcome measures except for MH and SF domains of SF-36. S3 Table provides descrip-tive statistics and pairwise post-hoc comparisons of all outcome measures.

### Primary outcome: Balance (mini-BEST)

At baseline, no significant between-group difference in mini-BEST scores was observed. Compared to the baseline, the mini-BEST scores of all three groups at post-treatment were not significantly different (S3 Table). For the DB group, the mini-BEST score at follow-up was significantly greater (p = 0.04) than the baseline. For the DB + DM group, the mini-BEST

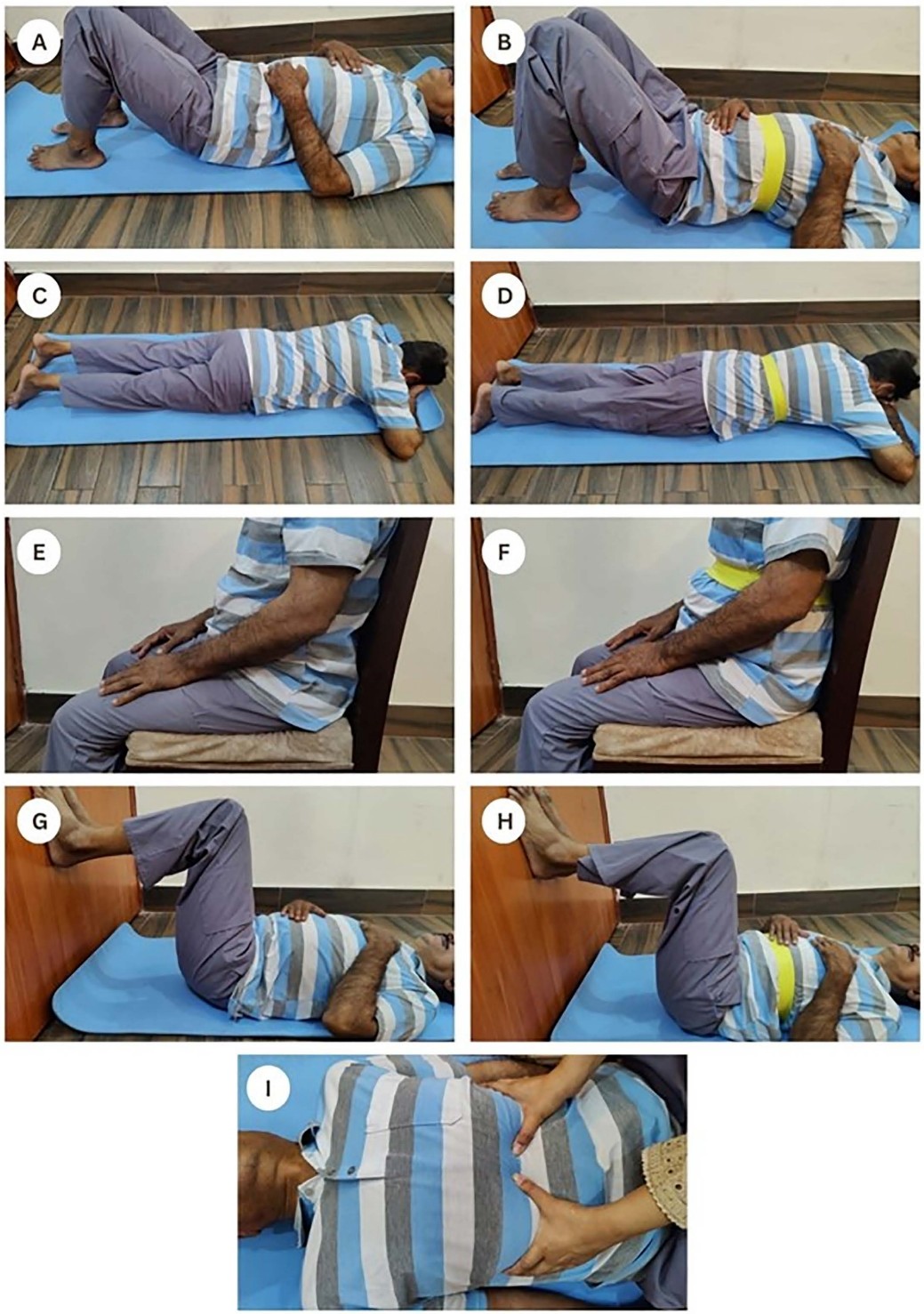

**Fig 3. Progression protocol of diaphragmatic breathing for both intervention groups A; Supine breathing without TheraBand, B; Supine breathing with TheraBand, C; Crocodile breathing without TheraBand, D; Crocodile breathing with TheraBand, E; Seated breathing without TheraBand, F; Seated breathing with TheraBand, G; 90/90/90 breathing without TheraBand, H; 90/90/90 breathing with TheraBand, I; Diaphragmatic mobilization (DM).**

score at follow-up was greater (p = 0.02) than the baseline. In contrast, the mini-BEST score of the control group significantly decreased from baseline to post-treatment (p = 0.02) and from baseline to follow-up (p < 0.001). When compared between groups, the mini-BEST score of the DB + DM group was significantly higher (p = 0.01) than the control group at post-treatment. At follow-up, the mini-BEST scores of the DB (p = 0.003) and DB + DM (p < 0.001) were significantly greater than the control group (S3 Table). The **between-group main effect** was significant (p = 0.010, η² = 0.164), and the **group × time interaction** was also significant (p < 0.001, η² = 0.389), indicating that changes over time differed across groups (S2 Table).

### Secondary outcomes

**Gait performance (TUG and GV).** At baseline, no significant differences in TUG scores were observed between groups. For the DB group, the TUG scores at post-treatment (p = 0.017) and follow-up (p = 0.005) were significantly lower than those at baseline (S3 Table). Likewise, the TUG scores of the DB + DM group at post-treatment (p = 0.02) and follow-up (p = 0.007) were significantly lower than that at baseline. Additionally, the TUG score at follow-up was significantly lower than at post-treatment (p = 0.03). In contrast, there was no significant difference in the TUG scores of the control group among the three measurements. When compared between the groups, the TUG score of the DB + DM group was marginally lower than that of the control group (p = 0.05) at post-treatment. However, at follow-up, the TUG score of the DB + DM group was significantly lower than the control group (p = 0.02).

For the GV, no significant between-group differences were observed at baseline. For the DB and DB + DM groups, the GV at post-treatment and follow-up were significantly faster than at baseline (p < 0.001) (S3 Table). Additionally, the GV at follow-up of both groups was significantly faster than at post-treatment (p < 0.001). In contrast, there was no significant difference in the GV of the control group among the three measurements. As a result, the GV of both intervention groups at post-treatment and follow-up were significantly greater than those of the control group measured at the same time point (p < 0.001). These results indicated the positive effect of DB and DB + DM on GV in community-dwelling older adults.

The group × time interaction was significant for TUG (p < 0.001, η² = 0.279), indicating that changes over time differed across groups. For GV, both the between-group main effect (p < 0.001, η² = 0.580) and the group × time interaction (p < 0.001, η² = 0.619) were significant, indicating differential changes over time between groups (S2 Table).

**Lower extremity strength (5xSTS).** Although a significant group-by-time interaction effect on 5xSTS performance was observed (p = 0.001, η² = 0.163) as shown in S2 Table, no pairwise significant differences were identified at any time point across the groups (S3 Table).

**Fear of falling (ABC scale).** At baseline, no significant differences in the ABC scores were observed between the three groups. For the DB group, the ABC scores at post-treatment and follow-up were significantly greater than at baseline (p < 0.001). Likewise, the ABC scores of the DB + DM group at post-treatment and follow-up were also significantly greater than at baseline (p < 0.001). In contrast, the ABC scores of the control group significantly decreased from baseline to post-treatment (p < 0.001), from post-treatment to follow-up (p = 0.01), and from baseline to follow-up (p < 0.001). As a result, there were significant differences in the ABC scores between the intervention groups and the control group at post-treatment (p < 0.001) and follow-up (p < 0.001) (S3 Table). These findings indicated that the DB and DB + DM groups significantly improved FoF than the control group.

The between-group main effect was significant (p < 0.001, η² = 0.501), and the group × time interaction was also significant (p < 0.001, η² = 0.706), indicating that changes over time differed across groups (S2 Table).

**Fatigue (FSS).** At baseline, FSS scores differed significantly between groups, with the DB (p = 0.007) and DB + DM (p = 0.002) groups showing lower fatigue levels than the control group (S3 Table). These baseline differences were taken into account when interpreting the results, with subsequent analyses focusing on changes over time and the group × time interaction. Within-group analyses showed significant reductions in FSS scores in the DB and DB + DM groups from baseline to post-treatment (p < 0.001) and from baseline to follow-up (p < 0.001). In contrast, no significant changes in

FSS scores were observed between the three measurements of the control group (S3 Table). Significant between-group differences in the FSS score were noted at post-treatment and follow-up. At post-treatment and follow-up, the FSS scores of the DB and DB+DM groups were significantly greater (p<0.001) than the control group. Moreover, at post-treatment, the FSS score of the DB+DM was significantly lower than the DB group (p=0.04). At follow-up, the FSS score of the DB+DM was marginally lower than the DB group (p=0.05). These findings indicate that both intervention groups significantly reduced fatigue compared to the control group.

The between-group main effect was significant (p<0.001, η²=0.636), and the group×time interaction was also significant (p<0.001, η²=0.584), indicating that changes over time differed across groups (S2 Table).

**Quality of life (SF-36).** The significant group-by-time interaction effects for several SF-36 domains, including PF (p=0.002, η²=0.211); RLP (p=0.041, η²=0.092), RLE (p=0.032, η²=0.097), energy (p=0.005, η²=0.136), pain (p=0.008, η²=0.173), and GH (p=0.008, η²=0.168) (S2 Table). However, pairwise comparisons with post-hoc analysis with Bonferroni adjustments showed significant pairwise comparisons for all domains (S3 Table). At baseline, there were no significant between-group differences in the scores of all domains except for the MH and GH domains. No significant within-group changes in the scores of all domains from baseline to post-treatment to follow-up were noted in the control group. In the DB and DB+DM groups, the PF, MH, Pain, and GH scores significantly increased from baseline to post-treatment and from baseline to follow-up (p<0.05). In the DB+DM group, PF, MH, and pain scores significantly increased from baseline to post-treatment and from baseline to follow-up (p<0.05). As a result, the PF, MH, and GH scores of the DB and DB+DM groups were significantly greater than those of the control group at post-treatment and follow-up. In addition, the RLP, energy, MH, pain, and GH scores of the DB+DM group were significantly greater than those of the control group at post-treatment and follow-up (S3 Table). These results indicated both intervention groups significantly improved QoL compared to the control group.

**Adverse events.** Both groups reported no adverse events or cardiovascular issues throughout the intervention period. BP remained stable before and after sessions across all participants. RPE, assessed using the Borg scale, consistently ranged from 11 to 13, indicating moderate intensity. On average, a one-minute rest between sets was sufficient to maintain the safety and tolerability of the interventions.

## Discussion

The purpose of this study was to preliminarily investigate the effects of DB and DB+DM on balance, gait performance, LES, FoF, fatigue, and QoL in community-dwelling older adults. To our knowledge, this study is the first to identify the potential effects of these two interventions related to functional mobility of balance and gait and overall well-being in older adults. Our results demonstrated that DB and DB+DM significantly improved gait velocity, FoF, fatigue, and QoL compared to no intervention. However, based on the results at post-treatment, these two interventions demonstrated no effects on balance, TUG, and LES in our participants. These results suggest that DB and DB+DM may improve certain aspects of physical performance and well-being when combined with additional or more targeted interventions.

Reducing the risks of falling, including impaired balance and gait, is one of the ultimate goals for fall prevention to maintain functional independence in older adults [43]. Previous studies have shown that DB interventions can improve balance in healthy adults by reducing sway velocity during static standing [44] and reducing errors during single-leg stance and tandem stance tasks [20]. Likewise, DB combined with transcutaneous electric nerve stimulation improved static and dynamic balance in athletes with chronic low back pain [22]. This study used the mini-BEST test to identify balance in community-dwelling older adults. Although both intervention groups demonstrated increased mini-BEST scores compared to the control group, these improvements were not statistically significant at post-treatment and only reached significance at follow-up. One likely explanation for the absence of early improvements is that DB and DM primarily enhance diaphragmatic activation, trunk proprioception, and postural alignment [19,20] rather than providing the multisystem challenge required to improve more complex balance tasks captured by the mini-BESTest. Thus, the benefits of improved

trunk control may require more time to translate into functional balance changes. Despite this statistical significance at follow-up, changes did not exceed the 4-point MDC, indicating limited clinical significance [45]. In contrast, the control group significantly declined the mini-BEST scores from baseline to follow-up. As a result, a significant difference in the mini-BEST scores between the DB + DM and control group was observed. Since the mini-BEST scores of our participants at baseline were consistent with those with a history of falling [46], these results underscore the therapeutic effects of these two interventions in maintaining and potentially improving balance in older adults at risk of falling. These results may suggest that these interventions could yield clinically significant results if implemented with a longer duration or higher-intensity training protocol.

In addition to balance impairment, gait impairment is also one of the risks of falling in older adults. In fact, gait speed and TUG tests have been recommended as screening tools for fall risk in older adults [47]. In our study, both DB and DB + DM groups showed improved gait performance, demonstrated by reduced TUG times and increased GV from baseline to post-treatment and follow-up. These findings align with previous studies that reported reduced TUG times after participating in DB in older females capable of independently performing daily activities [48]. While in our study, TUG improvements in both groups were statistically significant, they did not exceed the MDC of 1.4 seconds [49] nor surpass the fall-risk cutoff of 13.5 seconds. In contrast, gains in GV exceeded the fall-risk threshold of 0.8 m/s for older adults [50]. These findings suggest the potential of DB and DB + DM interventions to improve gait performance in older adults. Although the observed changes in TUG scores were small, such improvements, combined with enhanced GV, may reduce fall risk and enhance functional independence in older adults.

LES is one of the important factors in maintaining functional independence and mobility in older adults, as its decline is strongly associated with an increased risk of falls [51]. In our study, neither the DB nor DB + DM groups showed significant improvements in LES, as measured by the 5xSTS. Although a significant group × time interaction was observed for 5xSTS, the absence of pairwise differences indicates that these changes were not clinically meaningful in terms of lower extremity strength improvements. These results contradict the effects of breathing exercises in young adults previously reported [52,53]. For example, bracing breathing was shown to significantly improve postural stability and LES in healthy young males [52]. Similarly, DB alone and DB combined with lower extremity exercises significantly enhanced LES and balance in healthy young adults [53]. The differences between our results and previous studies may be attributable to a few factors. First, bracing breathing is a more intense exercise targeting core and lower limb muscle function, thereby providing a mechanical load sufficient to stimulate muscle strength [54]. In contrast, DB and DM emphasize spinal mobility and postural stability and do not have a direct impact on lower extremity strengthening. As a result, these two interventions were unlikely to elicit measurable changes in 5xSTS performance in our older adult participants. Additionally, between-study differences may play a role in differences in the results, as older adults due to age-related physiological changes [55] may require additional or more intensive strength-specific exercises to achieve comparable improvements in LES, unlike younger, healthy individuals.

FoF is one of the psychological factors contributing to fall risk, significantly impacting balance confidence and overall functional independence [56,57]. This study used the ABC Scale to assess FoF [58]. The significant improvement in the ABC scores from baseline to post-treatment and follow-up and significantly greater ABC scores between the two interventions and the control group highlight the effectiveness of these two interventions in reducing FoF. These findings are consistent with previous research where dynamic neuromuscular stabilization exercises with DB improved balance and reduced FoF in older females with a history of falls [59]. The reduction in FoF resulting from these two interventions will likely positively influence participation in physical activities and daily tasks, thereby reducing fall risk.

Fatigue is a significant concern in older adults, as it can lead to decreased physical activity, reduced participation in daily activities, and an increased risk of falling [60]. In this study, fatigue levels, as indicated by the FSS score, were significantly lower at baseline in the DB and DB + DM groups compared to the control group. This between-group baseline imbalance was considered when interpreting the observed changes. Both intervention groups demonstrated further

significant reductions in fatigue levels post-treatment and at follow-up. In contrast, no statistically significant changes in fatigue levels were observed in the control group, underscoring the specificity of the interventions. These findings align with previous research where DB significantly reduced fatigue in registered nurses [61]. Reducing fatigue through these interventions can enhance physical and psychological well-being, supporting greater independence and lowering the risk of falls in older adults.

QoL is one of the important components of fall prevention strategies, as it encompasses the physical and psychological well-being of older adults [62,63]. Poor QoL in older adults is often associated with diminished physical performance, contributing to a higher risk of falling [64]. In our study, both intervention groups showed significant improvements in QoL, consistent with prior research demonstrating enhanced QoL through DB in independently living older females [48]. Improvements in SF-36 subdomains, such as MH, pain, and GH, were observed in both the DB and DB + DM groups, aligning with findings from studies on DB and DM in patients with shoulder pain [28]. While the shoulder pain study highlighted improvements in role limits due to emotional problems, our study showed specific improvements in the DB group for the PF and in the DB + DM group for the RLP and energy domains. By reducing fatigue and enhancing QoL, DB and DB + DM interventions can promote functional independence, decrease fall risk, and improve the overall health and well-being of older adults.

The overall premise of this study is that improving control of the trunk via DB or in combination with DM would improve overall balance and gait performance. Due to its large mass compared to other body parts, the trunk plays a crucial role in postural stability. Targeted interventions addressing spinal posture have been shown to improve postural control, physical function, and quality of life in older adults [65]. Previous studies have demonstrated that DB and DM target the diaphragm, a muscle crucial for both respiration and postural stability [66]. DB enhances postural control by improving intra-abdominal pressure regulation and proprioception, contributing to better balance [19,20]. DM complements these effects by increasing thoracic and spinal mobility [24], facilitating optimal diaphragm function, and integrating sensory input for postural control [25]. Based on the results of our study, although both DB and DB + DM significantly improved gait performance, as indicated by the significant reductions in TUG times and improvement in GV scores, their impact on LES, as measured by 5xSTS, was not significant. This suggests that diaphragmatic exercises primarily target core stability rather than directly strengthening lower limb muscles. Trunk control is critical for maintaining postural stability [67], yet it works in conjunction with adequate LES to achieve optimal balance. The lack of significant improvements in LES may have limited the extent to which balance, as measured by the mini-BEST, could improve in this study. These findings underscore the importance of incorporating targeted lower extremity strengthening exercises into fall prevention programs to achieve more comprehensive improvements in balance and postural stability in older adults.

DB and DM are low-cost, non-invasive interventions that can be easily implemented in community and clinical settings. They require minimal equipment and can be taught by trained physiotherapists, making them feasible for older adults with mild to moderate functional limitations. Their low-intensity nature also allows integration with other rehabilitation or exercise programs, supporting adherence and safety in clinical practice.

This study has several limitations. As previously stated, the limited sample size and the twice-weekly 8-week intervention duration constrain the generalizability and long-term applicability of the findings. However, our participants were community-dwelling older adults with balance impairments. Therefore, our results will be applicable to those with balance impairment but not those with severe functional limitations. A further limitation is the baseline imbalance in FSS scores between groups, which may have contributed to variability in post-intervention outcomes despite statistical adjustment. A short 2-week follow-up may also not capture the long-term effects of the interventions and could reflect the carry-over effects. Future research should include longer follow-ups to assess sustainability. Another potential limitation is the possibility of a learning effect on performance-based measures such as the mini-BESTest, as repeated exposure to the test could influence participant scores. Although adherence among those who completed the study was high, we acknowledge

that adherence rates were not explicitly reported, and this information would provide additional context for interpreting the intervention's feasibility. Lastly, while the results demonstrated statistically significant improvements across most outcomes, the clinical significance of these changes, as reflected by the minimal clinically important difference (MCID) and minimal detectable change (MDC) thresholds, was not achieved for mini-BEST and TUG. Nonetheless, the modest improvements observed in balance scores are noteworthy, particularly given the baseline balance limitations seen in older adults. These small improvements in balance and gait performance suggest potential benefits for reducing fall risk over time. These findings highlight the importance of targeted, low-intensity interventions in addressing balance deficits, improving their FoF, and enhancing their QoL. Comparative trials integrating diaphragmatic techniques with other fall prevention strategies are recommended to optimize fall risk reduction. Additionally, modifications to intervention protocols, such as increasing duration, intensity, or incorporating resistance training, should be investigated to achieve more substantial improvements in this population, as the minimum effective dose of DB and DB + DM in older adults has yet to be established.

## Conclusion

This study demonstrates that DB and DB + DM statistically improved gait performance, fear of falling, fatigue, and quality of life in older adults. However, the improvements observed in balance and TUG did not reach the thresholds for clinical significance. Although some outcomes did not achieve the MCID/MDC thresholds, the observed improvements from baseline to follow-up underscore the potential of these interventions in fall prevention and in promoting functional and psychological well-being among older adults. These findings highlight the value of incorporating such interventions into targeted rehabilitation strategies for this population.

## Supporting information

**S1 Table. Demographic characteristics of participants.** This table summarizes the demographic characteristics of all participants across the three groups. No statistically significant differences were found in any demographic variable among the groups (p > 0.05), indicating that participants had similar prognostic factors at baseline.
(DOCX)

**S2 Table. Overall statistical analysis results of outcome measures.** This table presents the results of the group-by-time interaction for all primary and secondary outcome measures. A significant interaction effect was observed for all outcomes except for the Mental Health (MH) and Social Functioning (SF) domains of SF-36.
(DOCX)

**S3 Table. Descriptive statistics and pairwise post-hoc comparisons of outcome measures.** The table presents the mean and standard deviation at baseline, post-treatment, and follow-up, and includes within-group change from baseline (Δ) with 95% confidence intervals (95% CI), where 95% CI are shown as lower and upper bound values. It also includes post hoc pairwise comparisons for both within-group and between-group differences.
(DOCX)

**S1 File. Trial study protocol (approved by ethical committee).** The approved trial study protocol outlining the study design, participants, interventions, outcome measures, and ethical approval obtained from the Ethics Committee of Chulalongkorn University.
(DOCX)

**S2 File. CONSORT checklist.** The Consolidated Standards of Reporting Trials (CONSORT) checklist completed for this randomized controlled trial.
(DOCX)

## Author contributions

**Conceptualization:** Soyba Nazir, Witaya Mathiyakom, Anong Tantisuwat.

**Data curation:** Soyba Nazir, Muhammad Awais Tassawar.

**Formal analysis:** Soyba Nazir, Witaya Mathiyakom, Muhammad Awais Tassawar, Anong Tantisuwat.

**Investigation:** Anong Tantisuwat.

**Methodology:** Soyba Nazir, Witaya Mathiyakom, Anong Tantisuwat.

**Project administration:** Anong Tantisuwat.

**Supervision:** Witaya Mathiyakom, Anong Tantisuwat.

**Writing – original draft:** Soyba Nazir.

**Writing – review & editing:** Witaya Mathiyakom, Anong Tantisuwat.

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
