## [Decision Letter · Decision Letter 0]

3 Jul 2025

Dear Dr. Tantisuwat,

Thank you for submitting your manuscript to PLOS ONE. After careful consideration, we feel that it has merit but does not fully meet PLOS ONE’s publication criteria as it currently stands. Therefore, we invite you to submit a revised version of the manuscript that addresses the points raised during the review process.

We look forward to receiving your revised manuscript.

Kind regards,

Miray Budak

Academic Editor

PLOS ONE

Journal Requirements:

“NO authors have competing interests”

5. Please amend the manuscript submission data (via Edit Submission) to include author Tsuyoshi Asai.

6. Please amend your authorship list in your manuscript file to include author Muhammad Awais Tassawar.

Reviewers' comments:

Reviewer's Responses to Questions

**Comments to the Author**

1. Is the manuscript technically sound, and do the data support the conclusions?

Reviewer #1: Yes

Reviewer #2: Yes

Reviewer #3: Yes

2. Has the statistical analysis been performed appropriately and rigorously?

Reviewer #1: Yes

Reviewer #2: Yes

Reviewer #3: Yes

3. Have the authors made all data underlying the findings in their manuscript fully available?

Reviewer #1: Yes

Reviewer #2: No

Reviewer #3: Yes

4. Is the manuscript presented in an intelligible fashion and written in standard English?

Reviewer #1: Yes

Reviewer #2: Yes

Reviewer #3: Yes

Reviewer #1: The authors recruited 54 older adults to evaluate the effects of DB and DB+DM on physical and psychosocial outcomes. They observed improved gait performance, fear of falling, fatigue, and quality of life for both intervention groups.

1. Line 128. The sample size calculation is based on the effect size of 0.3. do you refer to standardized effect size? If yes, please update the text accordingly. Otherwise, what value of standard deviation was used?

2. Line 130. Sample size calculation yields 39 participants. Please be clear whether this is the total sample size across all three groups or for each group? Also, it should be clearly described what test was used for power analysis beyond just saying G*Power package.

3. Figure 1. Each group has two drop out from 18 participants but then 18 participants being analyzed, rather than 16. This may cause confusion.

Reviewer #2: Hi Dear Authors

why did not present results by charts and diagram?......................................................................................................................................

Reviewer #3: 1. Line number 315, 346, 363: Avoid the term elderly since it is considered as ageist. Older adults are a more acceptable term and I suggest to use the term older adults uniformly throughout the manuscript

2. Good study and well reported

3. Reference 58 and 59 are superscripted. Follow the journal guidelines closely.

**Do you want your identity to be public for this peer review?** For information about this choice, including consent withdrawal, please see our Privacy Policy

Reviewer #1: No

Reviewer #2: **Yes: ** soheil Mansour sohani

Reviewer #3: No

---

## [Author Response · Author response to Decision Letter 1]

16 Jul 2025

Dear Dr. Miray Budak and Reviewers,

Thank you for your constructive feedback and the opportunity to revise and improve our manuscript. We have thoroughly addressed all the comments from the editorial office and reviewers. Below is a point-by-point table listing each comment along with our response. All revisions are marked in the file titled “Revised Manuscript with Track Changes.” We hope these revisions meet your expectations, and we sincerely appreciate the time you took to provide us with your valuable feedback.

---

## [Decision Letter · Decision Letter 1]

18 Nov 2025

We look forward to receiving your revised manuscript.

Kind regards,

Miray Budak

Academic Editor

PLOS ONE

Journal Requirements:

Additional Editor Comments:

Thank you for submitting your manuscript to PLOS ONE. We invite you to submit a revised version of the manuscript that addresses the points raised during the review process.

Reviewer's Responses to Questions

**Comments to the Author**

Reviewer #1: All comments have been addressed

Reviewer #4: (No Response)

2. Is the manuscript technically sound, and do the data support the conclusions?

Reviewer #1: (No Response)

Reviewer #4: Yes

3. Has the statistical analysis been performed appropriately and rigorously?

Reviewer #1: (No Response)

Reviewer #4: Yes

4. Have the authors made all data underlying the findings in their manuscript fully available?

Reviewer #1: (No Response)

Reviewer #4: Yes

5. Is the manuscript presented in an intelligible fashion and written in standard English?

Reviewer #1: (No Response)

Reviewer #4: Yes

Reviewer #1: (No Response)

Reviewer #4: Dear Editor,

I have carefully reviewed the manuscript titled “The effect of diaphragmatic breathing and diaphragmatic mobilization on physical performance, fear of falling, and quality of life in older adults: a randomized controlled trial” submitted to PLOS ONE. I appreciate the authors’ effort in addressing the topic; however, I have several comments and concerns that I believe should be addressed before the manuscript can be considered for publication.

Minor revisions required. The manuscript has potential but requires substantial improvements in clarity, methodology reporting, statistical analysis, and interpretation of results.

Thank you for the opportunity to review this manuscript. I hope my comments are helpful in improving the quality and rigor of this work.

Sincerely

Title

• The title could be more precise and should indicate that the study focuses on community-dwelling older adults.

Abstract (lines 22–43)

• Lines 33–40: Effect sizes for significant results are not reported.

• Lines 26, 40: The difference between DB and DB+DM groups is unclear.

• Lines 41–42: The conclusion appears overly optimistic despite no improvement in post-treatment balance.

Introduction (lines 46–106)

• Lines 82–88: The distinct mechanism of DM compared to DB is not explained.

• Lines 74–78, 85–88: References to studies in younger adults are cited without caution regarding generalizability to older adults.

• Lines 101–104: The hypothesis is overly broad and lacks prioritization.

• The section is lengthy and includes excessive details about previous studies; it could be more concise.

• Some sentences are overly theoretical without direct references (e.g., the hydraulic effect of the diaphragm). Each claim should be supported by an appropriate reference.

The following studies can be used in your manuscript, with suggestions on where they might fit:

Naderi, A., Shaabani, F., Esmaeili, A., Salman, Z., Borella, E., & Degens, H. (2019). Effects of low and moderate acute resistance exercise on executive function in community-living older adults. Sport, Exercise, and Performance Psychology, 8(1), 106.

Can be cited in the Introduction when discussing the cognitive benefits of short-term resistance exercise in older adults, and in the Discussion when interpreting findings related to exercise and executive function.

Naderi, A., Aminian‐Far, A., Gholami, F., Mousavi, S. H., Saghari, M., & Howatson, G. (2021). Massage enhances recovery following exercise‐induced muscle damage in older adults. Scandinavian Journal of Medicine & Science in Sports, 31(3), 623-632.

Suitable for the Introduction when emphasizing non-pharmacological recovery strategies, and in the Discussion when addressing interventions to reduce exercise-induced fatigue in older adults.

Naderi, A., Goli, S., Shephard, R. J., & Degens, H. (2021). Six-month table tennis training improves body composition, bone health and physical performance in untrained older men; a randomized controlled trial. Science & Sports, 36(1), 72-e1.

Can be included in the Introduction to highlight the benefits of long-term, multidimensional training programs on body composition, bone health, and performance, and in the Discussion when interpreting improvements in physical performance outcomes.

Naderi, A., Rezvani, M. H., Shaabani, F., & Bagheri, S. (2019). Effect of kyphosis exercises on physical function, postural control and quality of life in elderly men with hyperkyphosis. Iranian Journal of Ageing, 13(4), 464-479.

propriate for the Introduction when introducing posture-specific exercise interventions for older adults, and in the Discussion when considering targeted interventions for kyphosis or postural control.

Methods – Study Design

• Line 153: The control group received no intervention, which may introduce attention bias.

• Lines 32, 186: Follow-up was only 2 weeks, which is insufficient.

• Line 116: Trial registration number is not reported in the main text.

Methods – Participants and Randomization

• Lines 129–130: Sample size calculated with medium effect size, but actual power is low.

• Line 127: Excluding participants with BMI >30 limits generalizability.

• Line 135: Random sequence generation details are missing.

• Line 133: The exact numbers of screened vs. eligible participants are not reported.

Methods – Intervention

• Lines 139, 145: Total intervention dose (320 minutes) is very low.

• Lines 145–150: Actual difference between DB and DB+DM is only 3 minutes of DM.

• Nowhere: Adherence rate is not reported.

• Nowhere: Number of missed sessions is not reported.

• Some intervention details (e.g., DM execution with figure-8 pattern) are overly technical and could be summarized in a figure or table.

• Follow-up duration of only 2 weeks limits assessment of long-term effects.

• Nowhere: Inter-rater reliability for mini-BEST is not reported.

Methods – Statistical Analysis

• Line 194: Using LOCF is outdated and may introduce bias.

• Lines 190–196, S3 Table: No adjustment for baseline imbalance in FSS.

• Only in S2 Table: Effect sizes (η²) are not reported in the main text.

Results (lines 197–282)

• Lines 251–252, S3 Table: Baseline imbalance in FSS undermines fatigue-related results.

• Lines 204–277: Numerical change from baseline (Δ) is not reported in the text.

• Lines 201–202: Claim of interaction without reporting η².

• Only in S3 Table: Exact p-values for pairwise comparisons not reported in the text.

• Nowhere: No mention of not reaching MDC in TUG/mini-BEST.

• Some results (e.g., 5xSTS) claim significance but pairwise analysis shows no difference; recommend clarifying statistical vs. clinical significance.

• Some p-values are borderline (e.g., p=0.05) and require cautious interpretation.

• Reporting of clinical effects (MDC, MCID) is limited; it should be provided for each main outcome.

Discussion (lines 284–412)

• Line 292: Claims of “promising effects” despite no post-treatment balance improvement.

• Lines 303–304: Interpretation of delayed effect in mini-BEST is not supported by evidence.

• Nowhere: Baseline imbalance in FSS is not discussed.

• Lines 331–339: Comparison with younger adult studies lacks caution.

• Nowhere: Cost, training requirements, and feasibility in clinical practice are not discussed.

• Lines 409–410: Suggestion to increase duration without defining minimum effective dose.

• Sections are long and repetitive, particularly regarding DB and DB+DM effects on TUG and GV.

• Explanation of null effects (e.g., 5xSTS and mini-BEST post-treatment) could be strengthened with physiological rationale or exercise intensity considerations.

• Discussion could be more concise with practical recommendations for clinical application and future research.

• Discussion of MCID/MDC limitations could be expanded to help readers distinguish statistical vs. clinical significance.

Tables and Figures

• S1 Table: Gender not reported by group.

• S3 Table: Δ from baseline column missing.

• S1 Fig: Inconsistency in number analyzed (CONSORT).

• S3 Fig caption: Insufficient explanation for “90/90/90 breathing.”

Limitations (lines 393–412)

• Nowhere: Baseline imbalance in FSS not mentioned as a limitation.

• Line 393: Low dose not identified as a primary limitation.

• Nowhere: Adherence rate not reported.

• Nowhere: Possible learning effect in mini-BEST not discussed.

Conclusion (lines 413–419)

• Line 414: Overstated claims of statistical improvement in balance and strength.

• Lines 415–416: No mention of the lack of clinical significance for balance and TUG.

References

• Line 528: Reference 35 (Cozean questionnaire) is from a non-official source.

• Lines 586, 593: References 58 and 59 formatted incorrectly.

Supplementary Information

• S2 Table: η² for group × time not reported in a separate column.

• S3 Table: 95% CI for changes not reported.

**Do you want your identity to be public for this peer review?** For information about this choice, including consent withdrawal, please see our Privacy Policy

Reviewer #1: No

Reviewer #4: No

---

## [Author Response · Author response to Decision Letter 2]

30 Nov 2025

Dear Dr. Miray Budak and Reviewers,

Thank you for your constructive feedback and the opportunity to revise and improve our manuscript. We have thoroughly addressed all the comments from the reviewer. Below is a point-by-point table listing each comment along with our response. All revisions are marked in the file titled “Revised Manuscript with Track Changes.” We hope these revisions meet your expectations, and we sincerely appreciate the time you took to provide us with your valuable feedback.

---

## [Decision Letter · Decision Letter 2]

14 Dec 2025

The effect of diaphragmatic breathing and diaphragmatic mobilization on physical performance, fear of falling, and quality of life in community-dwelling older adults: a randomized controlled trial

PONE-D-25-10162R2

Dear Dr. Tantisuwat,

We’re pleased to inform you that your manuscript has been judged scientifically suitable for publication and will be formally accepted for publication once it meets all outstanding technical requirements.

Kind regards,

Tanja Grubić Kezele, Ph.D., M.D.

Academic Editor

PLOS One

Additional Editor Comments (optional):

Reviewers' comments:

Reviewer's Responses to Questions

**Comments to the Author**

Reviewer #1: (No Response)

Reviewer #4: All comments have been addressed

2. Is the manuscript technically sound, and do the data support the conclusions?

Reviewer #1: (No Response)

Reviewer #4: Yes

3. Has the statistical analysis been performed appropriately and rigorously?

Reviewer #1: (No Response)

Reviewer #4: Yes

4. Have the authors made all data underlying the findings in their manuscript fully available?

Reviewer #1: (No Response)

Reviewer #4: Yes

5. Is the manuscript presented in an intelligible fashion and written in standard English?

Reviewer #1: (No Response)

Reviewer #4: Yes

Reviewer #1: (No Response)

Reviewer #4: (No Response)

**Do you want your identity to be public for this peer review?** For information about this choice, including consent withdrawal, please see our Privacy Policy

Reviewer #1: No

Reviewer #4: No

---

## [Editor Report · Acceptance letter]

PONE-D-25-10162R2

PLOS One

Dear Dr. Tantisuwat,

I'm pleased to inform you that your manuscript has been deemed suitable for publication in PLOS One. Congratulations! Your manuscript is now being handed over to our production team.

Kind regards,

on behalf of

Prof. dr. Tanja Grubić Kezele

Academic Editor

PLOS One